**Investigation**

# Centromere-proximal suppression of meiotic crossovers in *Drosophila* is robust to changes in centromere number, repetitive DNA content, and centromere-clustering

Nila M. Pazhayam,[1] Leah K. Frazier (ID),[2] Jeff Sekelsky (ID) [1,3,4,]*

[1]Curriculum in Genetics and Molecular Biology, University of North Carolina at Chapel Hill, Chapel Hill, NC 27599, USA
[2]SURE-REU Program in Biological Mechanisms, University of North Carolina at Chapel Hill, Chapel Hill, NC 27599, USA
[3]Integrative Program for Biological and Genome Sciences, University of North Carolina at Chapel Hill, Chapel Hill, NC 27599, USA
[4]Department of Biology, University of North Carolina at Chapel Hill, Chapel Hill, NC 27599, USA

*Corresponding author: Department of Biology, University of North Carolina at Chapel Hill, Chapel Hill, NC 27599, USA. Email: sekelsky@unc.edu

Accurate segregation of homologous chromosomes during meiosis depends on both the presence and the regulated placement of crossovers (COs). The centromere effect, or CO exclusion in pericentromeric regions of the chromosome, is a meiotic CO patterning phenomenon that helps prevent nondisjunction, thereby protecting against chromosomal disorders and other meiotic defects. Despite being identified nearly a century ago, the mechanisms behind this fundamental cellular process remain unknown, with most studies of the *Drosophila* centromere effect focusing on local influences of the centromere and pericentric heterochromatin. In this study, we sought to investigate whether dosage changes in centromere number and repetitive DNA content affect the strength of the centromere effect, using phenotypic recombination mapping. Additionally, we studied the effects of repetitive DNA function on centromere effect strength using satellite DNA–binding protein mutants displaying defective centromere-clustering in meiotic nuclei. Despite what previous studies suggest, our results show that the *Drosophila* centromere effect is robust to changes in centromere number, repetitive DNA content, as well as repetitive DNA function. Our study suggests that the centromere effect is unlikely to be spatially controlled, providing novel insight into the mechanisms behind the *Drosophila* centromere effect.

Keywords: centromere; meiosis; crossover; satellite DNA; heterochromatin

## Introduction

Meiosis is a specialized form of cell division in which a diploid chromosome set is reduced to a haploid set through the segregation of homologous chromosomes during anaphase of meiosis I (MI). Accurate disjunction, or segregation, is largely facilitated by crossing over between homologs, an integral part of MI that requires the formation of programmed double-strand breaks (DSBs) along the chromosome and involves a reciprocal exchange of chromosome arms. DSBs are repaired through homologous recombination to generate either crossover (CO) or non-CO (NCO) repair products. The decision of whether a DSB is repaired as a CO or NCO is a highly regulated process, as the frequency and positioning of meiotic COs is critical for proper segregation of homologs.

A far greater number of DSBs are formed during meiosis than are repaired as COs, and the mechanisms that regulate CO placement along the chromosome are collectively referred to as CO patterning phenomena. Three important patterning events are as follows: (1) assurance, which dictates that every pair of homologs receives at least one CO (Owen 1950); (2) interference, which ensures that COs do not form right next to one another (Sturtevant 1913); and (3) the centromere effect (CE), which dictates that COs are excluded from centromere-proximal regions (Beadle

1932). These have been reviewed extensively by Pazhayam *et al.* (2021). These phenomena are critical for accurate disjunction of homologous chromosomes in humans, safeguarding against miscarriages and chromosomal disorders.

The centromere effect is vital for preventing nondisjunction (NDJ), the risk of which increases with increasing maternal age. Studies in both humans and *Drosophila melanogaster* have established a direct link between pericentromeric crossing over and NDJ (Koehler *et al.* 1996; Lamb *et al.* 1996). Furthermore, human studies have established that missegregation events on chromosome 21, the leading cause of Down syndrome, correlate with centromere-proximal COs and increase with maternal age (Oliver *et al.* 2012). However, very little is known about the mechanisms behind the centromere effect, despite it being a critical safeguard against chromosome missegregation and age-associated meiotic defects.

The centromere effect was first reported in 1932 when Beadle observed a decrease in CO frequencies in *D. melanogaster* translocation stocks that had a portion of the 3rd chromosome moved closer to the centromere of the 4th than it had originally been to the centromere of the 3rd (Beadle 1932). He attributed this reduction in recombination rate to the chromosomal interval's increased proximity to the chromosome 4 centromere. Although first described over 9 decades ago and observed widely across

species (Mahtani and Willard 1998; Copenhaver *et al.* 1999; Vincenten *et al.* 2015; Nambiar and Smith 2016; Fernandes *et al.* 2023), the mechanisms behind the centromere effect have remained elusive to this day.

In most organisms, centromeres are surrounded by pericentromeric heterochromatin that consists of repetitive DNA, including large arrays of satellite DNA along with other repetitive elements. In *D. melanogaster*, nearly one-third of the genome is heterochromatic pericentromeric satellite DNA (Hoskins *et al.* 2002). A prominent question regarding the mechanism of the centromere effect has centered on the ways in which this heterochromatic, repetitive sequence and the centromere itself contribute to centromere-proximal CO suppression. Of the handful of studies that have addressed pericentromeric crossing over in the past century, most have attempted to establish one as more important than the other in *D. melanogaster*, with contrasting results. Heterochromatin has been considered everything from an active participant in CO reduction in adjacent intervals (Slatis 1955; John and King 1985; Westphal and Reuter 2002; Mehrotra and Mckim 2006) to nothing more than a passive spacer (Mather 1939; Yamamoto and Miklos 1978) between euchromatin and the centromere.

The influence of repetitive DNA on *D. melanogaster* CO frequencies has been studied in the past, however always in *cis*. A 1955 study measured COs in homozygous $bw^D$ mutants, flies with an ~2 Mb insertion of $\{AAGAG\}_n$ satellite sequence into the distal *brown* locus on chromosome 2R and observed a marked reduction in CO frequencies surrounding the insertion (Slatis 1955). The exclusion of DSBs in heterochromatic regions of flies, which are primarily repetitive, has also been shown (Mehrotra and Mckim 2006), suggesting that CO suppression in pericentromeric regions may be explained by decreases in DSB formation in the region. Decreased dosage of chromosome 4, which consists almost entirely of repetitive sequence, has also been shown to have *cis*-effects, reducing the expression of various chromosome 4 genes (Haynes *et al.* 2007).

Furthermore, repetitive DNA has been shown to have *trans*-effects on phenomena such as gene expression and heterochromatic integrity in *D. melanogaster*. In this study, we use the term *trans* to refer to effects on other chromosomes and not on the other homolog. Flies with extra copies of chromosome Y, which consists almost entirely of highly repetitive DNA, have been shown to derepress position effect variegation, the phenomenon where proximity to heterochromatin causes variegated gene expression (Muller 1930; Dimitri and Pisano 1989). This is thought to be due to the Y chromosome increasing competition for a limited pool of heterochromatic factors within a cell, leading to a decrease in heterochromatinization elsewhere in the genome, and the consequent derepression of genes close to heterochromatic boundaries. Several recent studies support this proposal, with lowered dimethylation and trimethylation of H3K9, a canonical mark of heterochromatin, observed at the pericentromeres of all chromosomes in XXY and XYY flies, compared to XX and XY flies (Brown *et al.* 2020), and increasing Y chromosome lengths negatively correlating with gene silencing in *trans* (Delanoue *et al.* 2023). Despite the implications of these studies, a key piece of the puzzle that remains unanswered is whether repetitive DNA and heterochromatin exert *trans*-effects on meiotic CO frequencies, particularly near the centromere.

*Cis*-effects of the centromere on CO frequencies have also been established in *D. melanogaster*, the first instance of which was Beadle (1932) concluding that CO frequencies in euchromatic regions decrease when brought closer to a centromere. In his 1939 study, Mather (1939) further concluded that CO frequencies in

euchromatin depend on proximity to the centromere. Similarly, Yamamoto and Miklos (1978) moved euchromatic regions closer to the centromere by deleting pericentromeric heterochromatin, showing that CO frequencies in these regions negatively correlate with proximity to the centromere. In the 1930s, Helen Redfield measured COs in triploid *D. melanogaster* females and observed substantial increases in centromere-proximal CO frequencies, compared with diploids (Redfield 1930, 1932). Although this change in centromere effect strength may be a result of ploidy changes, it is also possible that it is a consequence of the total number of centromeres in triploids increasing 1.5-fold, raising key questions about whether dosage changes in just centromere number are capable of exerting *trans*-effects on centromere-proximal CO frequencies.

To fill this gap in knowledge regarding the mechanisms of a fundamental cellular process, we measured centromere-proximal CO frequencies and centromere effect strength in mutants with dosage changes in centromere number and repetitive DNA content, respectively. We also measured centromere-proximal CO frequencies in mutants of satellite DNA–binding proteins that display significant centromere declustering in meiotic nuclei to ask whether repetitive DNA function plays a role in the establishment of the centromere effect. Surprisingly, our results show no change in the strength of the centromere effect in mutants with a decreased total number of centromeres, in mutants with increased and decreased total repetitive DNA content, or in satellite DNA–binding protein mutants that cause defective centromere-clustering during meiosis. Overall, our study suggests that the centromere effect is robust to dosage changes in centromere number and the amount of repetitive DNA, as well as certain aspects of repetitive DNA function.

## Materials and methods
### *Drosophila* stocks
Flies were maintained on a standard medium at 25°C. The following fly strains were obtained from the Bloomington Stock Center (NIH P40OD018537): 1612 [$C(1)RM$, $y^1$ $v^1/C(1;Y)^1$, $v^1$ $f^1$ $Bar^1/0$; $C(4)RM$, $ci^1$ $ey^R/0$], 9460 [$C(1)RM/C(1;Y)6$, $y^1$ $w^1$ $f^1/0$], 1785 [$C(4)RM$, $ci^1$ $ey^R/0$], and 741 [$Df(2R)M41A10/SM1$]. The following fly strains were generously gifted to us by Dr. Yukiko Yamashita: $prod^{K08810}/CyO$, $Act::GFP$ (referred to as $prod^K/+$ in the manuscript), $w$; $Df(3R)BSC^{666}/TM6C$, $Sb$, $y$ $w$ ; $D1^{LL03310}/TM6B$, $Tb$. Wild-type controls were *Oregon-R*, which was gifted to us by Dr. Scott Hawley.

### Genetic assays
Chromosome 2 COs were mapped by crossing virgin *net dpp^{ho} dp b pr cn*/+ females of desired mutant backgrounds to males that were homozygous *net dpp^{ho} dp b pr cn*. Chromosome 3 COs were mapped by crossing virgin *ru h th st cu sr e ca*/+ females of desired mutant backgrounds to males that were homozygous *ru h th st cu sr e ca*. Vials were set up with 1–5-day-old females and then flipped a week later. For both chromosomes 2 and 3, progenies were scored for all markers.

### Fly crosses
Compound females [$C(1)RM$, $y^1$ $v^1/0$; $C(4)RM$, $ci^1$ $ey^R/0$, $C(1)RM/0$, $C(4)RM$, $ci^1$ $ey^R/0$] were crossed to males homozygous for recessive markers on chromosome 2 or 3 to obtain XXY, triplo-4 [$C(1)RM$, $y^1$ $v^1/Y$; $C(4)RM$, $ci^1$ $ey^R/+$], XXY [$C(1)RM/Y$], and triplo-4 [$C(4)RM$, $ci^1$ $ey^R/+$] females heterozygous for recessive markers.

## Recombination calculation

Genetic distance is calculated as $100 \times (r/n)$, where $n$ is the total scored progeny and $r$ is the total recombinant progeny within an interval (including single, double, and triple COs). It is expressed in centiMorgans (cM). Variance was used to calculate 95% confidence intervals, as in Stevens (1936). Physical distances between recessive markers used for phenotypic CO mapping were calculated using positions of genetic markers on release 6.53 of the *D. melanogaster* reference genome. Distances were calculated from the beginning of a genetic marker to the end of the previous marker. Centromere effect (CE) values were calculated as 1–(observed COs/expected COs), where expected COs are calculated as follows: total COs × (length of proximal interval/total length).

## Larval neuroblast chromosome spreads

Brains were dissected from wandering 3rd instar larvae in cold PBS, incubated in 0.5% sodium citrate for 10 min, and then fixed in 2% formaldehyde and 45% acetic acid solution for 7 min on the siliconized cover slips. The brains were squashed onto glass slides (VWR Micro Slides, Superfrost Plus) and frozen in liquid nitrogen and then washed thrice in 0.1% Tween-20 in PBS (PBS-T) for 10 min each. The slides were blocked for 1 h in 1% BSA, PBS-T, solution and incubated overnight with an anti–CENP-C antibody (1:5,000 from Dr. Kim McKim) at 4°C. They were then washed 4 times in PBS-T for 10 min each and incubated for 2 h in a secondary antibody (1:500, antirabbit) at room temperature. The slides were washed 4 times again in PBS-T for 10 min each and then mounted with a 1:1,000 solution of 1 mg/mL DAPI in fluoromount-G.

## Immunofluorescence of germaria

Adult females of the desired genotype were fattened overnight at 25°C in vials containing yeast paste and males of any genotype at an ~3:1 female-to-male ratio. Ovaries were dissected in freshly prepared 1× PBS and then fixed for 20 min in a buffer consisting of 165 μL of freshly prepared 1× PBS, 600 μL of heptane, 25 μL of 16% formaldehyde, and 10 μL of N-P40. The ovaries were washed in 0.1% Tween-20 in PBS (PBS-T) thrice for 15 min each, blocked for 1 h in 1% BSA, PBS-T, and then incubated overnight at 4°C in a 1:1,000 dilution of anti-C(3)G antibody (from Dr. Nicole Crown) and a 1:5,000 dilution of anti–CENP-C antibody (from Dr. Kim McKim). The ovaries were washed thrice more in 1% PBS-T for 15 min and incubated for 2 h at room temperature in a 1:500 dilution of secondary antibodies [antimouse for C(3)G and antirabbit for CENP-C]. Five microliters of 1 mg/mL DAPI were added in the last 10 min of this incubation, following which the ovaries were washed again in 1% PBS-T, 3 times for 15 min each. They were then mounted on slides with 30 μL of fluoromount-G.

## Imaging

Chromosome spreads of larval brains were imaged using a Zeiss LSM880 confocal laser scanning microscope with Airyscan and also with the 63× oil immersion objective lens. FIJI (ImageJ) was used to process the images. Centromere clusters were quantified by manually counting CENP-C foci that colocalized with the DAPI-dense regions of C(3)G positive (meiotic) cells in adult germaria. Percent declustering was calculated as follows: (number of cells with >3 CENP-C foci colocalizing with DAPI-dense regions/total number of cells) × 100. Numbers of foci for each image are given in Supplementary Table 3.

# Results

## Centromere dosage does not exert *trans*-effects on the centromere effect

While the centromere's local, cis-acting effect on CO frequencies in centromere-proximal chromosomal regions has been established in *D. melanogaster* (Yamamoto and Miklos 1978), whether centromeres also exert *trans*-effects on CO frequencies remains unknown. Although Redfield (1930, 1932) showed that triploid *D. melanogaster* females have an increased number of centromere-proximal COs on chromosomes 2 and 3 compared with diploids, it is unclear whether this effect is due to a change in the total centromere number from 16 in diploids (which have 8 chromosomes with 1 centromere per sister chromatid) to 24 in triploids (which have 12 chromosomes with 1 centromere per sister chromatid) or a consequence of ploidy changes. We hypothesized that since Redfield's experiments showed a weakened centromere effect with an increase in centromere numbers, a reduction in centromeres would lead to a strengthened centromere effect. To investigate the effects of reducing total centromere number without the complicating factor of ploidy change, we made use of a *D. melanogaster* stock with compound *X* and *4th* chromosomes, henceforth referred to as *C(1)/0*; *C(4)/0*. Flies of this genotype have attached homologs of chromosome *X* as well as homologs of chromosome 4, with each attached chromosome having only 2 centromeres, thereby reducing the total centromere number from 16 to 12 (Fig. 1a).

To confirm this reduced centromere number, we performed CENP-C immunofluorescent staining on mitotic chromosome spreads of *C(1)/0*; *C(4)/0* flies. CENP-C is a part of the inner kinetochore and is widely used as a centromeric mark in chromosome spreads instead of CENP-A (Palladino *et al.* 2020), as the acetic acid fixative required for these spreads tends to degrade histones. Figure 1b shows larval neuroblast spreads and the 12 CENP-C foci that we observed in the *C(1)/0*; *C(4)/0* stock. This is a 25% reduction in centromere number from wild-type flies, where 16 CENP-C foci were observed (Fig. 1a).

To test whether a reduced centromere number in the *X* and *4th* chromosomes has a *trans*-effect on proximal CO frequencies, we mapped recombination on chromosome 2 through the phenotypic scoring of recessive markers distributed along it. CO densities along the left and right arms of chromosome 2—between the loci *net* (at the distal end of chromosome 2L) and *cinnabar* (situated 7.7 Mb into the assembled portion of chromosome 2R)—were measured using flies heterozygous for these markers. This includes all of chromosome 2's left arm and extends past the centromere into the right arm, covering a total of 31 Mb on chromosome 2, with the centromere located in the 11.2-Mb-long interval between markers *purple* and *cinnabar*, based on release 6.53 of the *D. melanogaster* reference genome.

Figure 1b shows CO density in the *C(1)/0*; *C(4)/0* mutant plotted across the 31-Mb-long region of chromosome 2, divided by recessive markers into 5 intervals. We were surprised to observe no change in CO frequencies within the *purple–cinnabar* interval, which contains the centromere. Reducing the centromere number from 16 to 12 did not produce a change in the density of proximal COs, implying that a reduced number of total centromeres does not exert a *trans*-effect on centromere-proximal CO suppression.

Since a distribution graph only compares observed CO frequencies in each interval between mutants and wild-type controls, we sought to also calculate a more biologically relevant measure of centromere effect strength that considers expected vs observed outcomes. This CE value acknowledges that an interval's expected

**(a)**

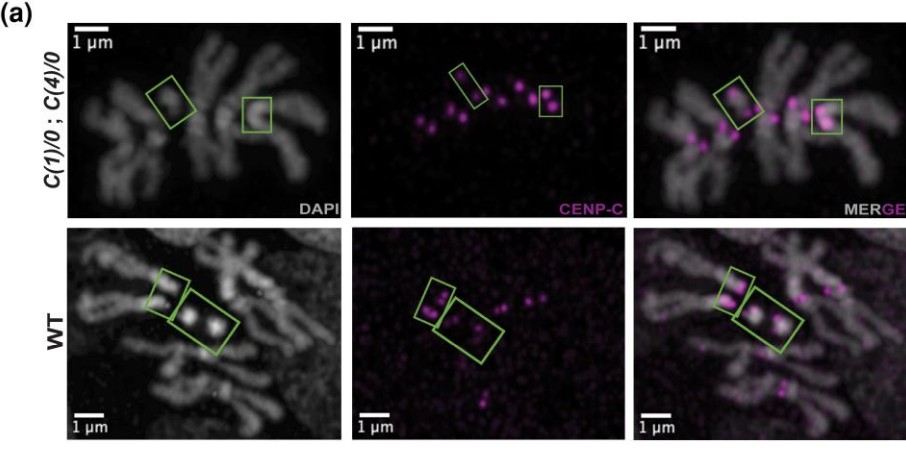

**(b)**

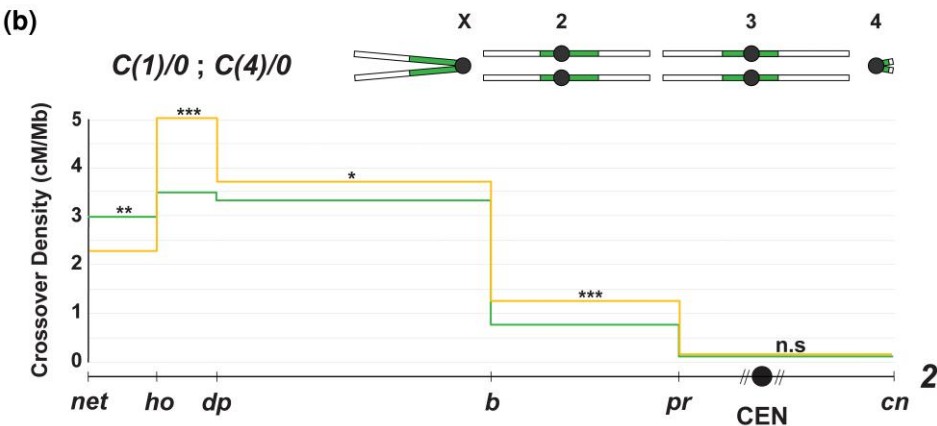

**(c)**

$$CE \ value = 1 - \frac{observed \ proximal \ COs}{expected \ proximal \ COs}$$

$$expected \ proximal \ COs = total \ COs * \frac{proximal \ interval \ length}{total \ length}$$

| Genotype | No. of CENs | Chr. *2* CE value |
|---|---|---|
| WT | 16 | 0.92 |
| *C(1)/0 ; C(4)/0* | 12 | 0.90 |

**Fig. 1.** a) Mitotic chromosome spreads (gray) with CENP-C foci (magenta) from *C(1)/0; C(4)/0* larvae (top panel) and wild-type larvae (bottom panel). The green boxes highlight chromosomes *X* (larger, V-shaped structures) and *4* (smaller, dot-like structures) in both genotypes. b) CO distribution along chromosome *2* in *C(1)/0; C(4)/0* flies (yellow line, *n* = 5,311) and wild-type flies (green line, *n* = 4,104), with a schematic of the *C(1)/0; C(4)/0* karyotype shown above the graph. CO density in cM/Mb is indicated on the *y*-axis, and relative physical distances between markers used to score COs are indicated on the *x*-axis. The black circle represents the centromere, with the dashed lines around it representing a pericentromeric repetitive sequence that remains unassembled. Statistical significance in each interval was calculated using a 2-tailed Fisher's exact test between the difference in total CO vs NCO numbers in mutants from wild-type flies (ns P > 0.0083, *P < 0.0083, **P < 0.0017, ***P < 0.00017 after correction for multiple comparisons). The complete dataset can be found in Supplementary Table 1. c) A mathematical definition of the CE value and table containing centromere number as well as the chromosome 2 CE values of WT and *C(1)/0; C(4)/0* flies. A 2-tailed Fisher's test was used to calculate the significance between observed and expected proximal CO values, and it was nonsignificant between *C(1)/0; C(4)/0* and wild-type flies.

CO numbers will depend on a genotype's total CO numbers across the chromosome, thereby accounting for differences in overall CO frequencies between genotypes (Fig. 1c). It allows us to compare observed CO frequencies with those expected were the centromere effect not regulating recombination rates near the centromere. When calculated for the *C(1)/0; C(4)/0* flies, CE on chromosome 2 was 0.90 (Fig. 1c). This is not significantly different from the wild-type CE of 0.92 on chromosome 2, further suggesting that dosage changes in centromere number do not have a *trans*-influence on the strength of the centromere effect.

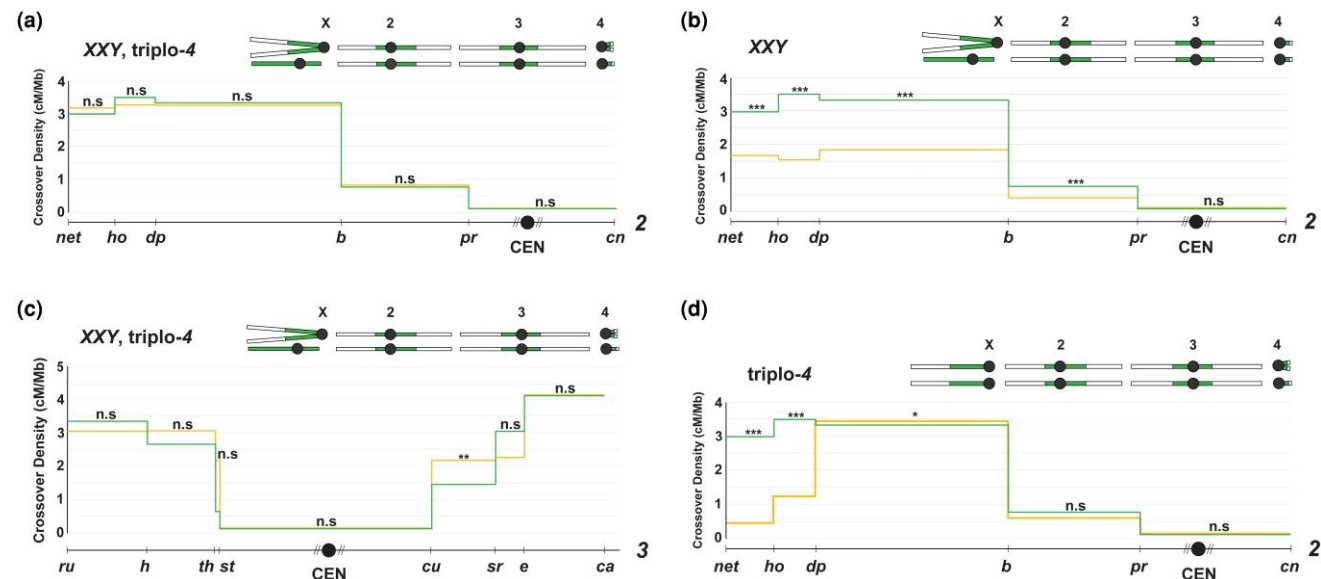

**Fig. 2.** a) CO distribution along chromosome 2 in *XXY*, triplo-4 flies (yellow line, *n* = 4,434) and wild-type flies (green line, *n* = 4,104), with a schematic of the *XXY*, triplo-4 karyotype shown above the graph. CO density in cM/Mb is indicated on the *y*-axis, and relative physical distances between markers used to score COs are indicated on the *x*-axis. The black circle represents the centromere, with the dashed lines around it representing the pericentromeric repetitive sequence that remains unassembled. Statistical significance in each interval was calculated using a 2-tailed Fisher's exact test between the difference in total CO vs NCO numbers in mutants from wild-type flies (ns *P* > 0.0083, *$^*P$* < 0.0083, *$^{**}P$* < 0.0017, *$^{***}P$* < 0.00017 after correction for multiple comparisons). The complete dataset can be found in Supplementary Table 1. b) CO distribution along chromosome 2 in *XXY* flies (yellow line, *n* = 3,787) and wild-type flies (green line, *n* = 4,104), with a schematic of the *XXY* karyotype shown above the graph. CO density in cM/Mb is indicated on the *y*-axis, and relative physical distances between markers used to score COs are indicated on the *x*-axis. The black circle represents the centromere, with the dashed lines around it representing pericentromeric repetitive sequence that remains unassembled. Statistical significance in each interval was calculated using a 2-tailed Fisher's exact test between the difference in total CO vs NCO numbers in mutants from wild-type flies (ns *P* > 0.0083, *$^*P$* < 0.0083, *$^{**}P$* < 0.0017, *$^{***}P$* < 0.00017 after correction for multiple comparisons). The complete dataset can be found in Supplementary Table 1. c) CO distribution along chromosome 3 in *XXY*, triplo-4 flies (yellow line, *n* = 922) and wild-type flies (green line, *n* = 1,728), with a schematic of the *XXY*, triplo-4 karyotype shown above the graph. CO density in cM/Mb is indicated on the *x*-axis, and relative physical distances between markers used to score COs are indicated on the *x*-axis. The black circle represents the centromere, with the dashed lines around it representing pericentromeric repetitive sequence that remains unassembled. Statistical significance in each interval was calculated using a 2-tailed Fisher's exact test between the difference in total CO vs NCO numbers in mutants from wild-type flies (ns *P* > 0.0125, *$^*P$* < 0.0125, *$^{**}P$* < 0.0025, *$^{***}P$* < 0.00025 after correction for multiple comparisons). The complete dataset can be found in Supplementary Table 2. d) CO distribution along chromosome 2 in triplo-4 flies (yellow line, *n* = 2,924) and wild-type flies (green line, *n* = 4,104), with a schematic of the triplo-4 karyotype shown above the graph. CO density in cM/Mb is indicated on the *y*-axis, and relative physical distances between markers used to score COs are indicated on the *x*-axis. The black circle represents the centromere, with the dashed lines around it representing pericentromeric repetitive sequence that remains unassembled. Statistical significance in each interval was calculated using a 2-tailed Fisher's exact test between the difference in total CO vs NCO numbers in mutants from wild-type flies (ns *P* > 0.0083, *$^*P$* < 0.0083, *$^{**}P$* < 0.0017, *$^{***}P$* < 0.00017 after correction for multiple comparisons). The complete dataset can be found in Supplementary Table 1.

## Increases in total repetitive DNA content do not exert *trans*-effects on the centromere effect

Pericentromeric heterochromatin and repetitive DNA can be thought of as exerting *cis*-effects on CO frequencies in 2 ways: first, through CO exclusion in highly repetitive, heterochromatic regions (Westphal and Reuter 2002; Mehrotra and Mckim 2006; Peng and Karpen 2009; Hartmann *et al.* 2019) and second, through suppressing COs in adjacent euchromatic, nonrepetitive regions (Slatis 1955; John and King 1985). In addition to *cis*-effects, repetitive DNA has also been shown in *D. melanogaster* to influence phenomena such as gene expression and heterochromatin integrity in *trans* (Dimitri and Pisano 1989; Brown *et al.* 2020; Delanoue *et al.* 2023). However, whether repetitive DNA and heterochromatin exert *trans*-effects on centromere-proximal CO frequencies is a question that remains unanswered.

To answer this question, we made use of flies with increased total amounts of repetitive DNA, starting with *XXY*, triplo-4 flies. The *D. melanogaster* chromosome 4 also consists almost entirely of repetitive sequence, and based on chromosome lengths and heterochromatic size estimates from the sequenced genome (Hoskins *et al.* 2002), triplo-4 flies—that have an extra copy of chromosome 4—have an ~3% increase in repetitive DNA content

compared with flies that are diplo-4 (Fig. 2d). Similarly, the Y chromosome in *XXY* females contributes to an ~35% increase in repetitive DNA content over wild-type *XX* females (Fig. 2b). Combined with the triplo-4 genotype, *XXY*, triplo-4 females have an ~37% increase in repetitive DNA content (due to additional copies of both chromosomes Y and 4) compared with *XX*, diplo-4 females (Fig. 2, a and c). We ensured that these mutants with increased repetitive DNA also did not have increased centromere numbers by using previously validated (Fig. 1a; Supplementary Fig. 1a) fly stocks with compound chromosomes of *X* and 4 to build them.

To test the *trans*-effects of increased repetitive DNA on centromere-proximal CO frequencies, we measured CO distribution across chromosomes 2 and 3 in these mutants, hypothesizing that the decreased heterochromatic integrity in *XXY* and *XYY* flies observed by Brown *et al.* (2020) would allow for greater centromere-proximal COs or a weaker centromere effect. Figure 2b shows CO density plotted across the 31 Mb between markers *net* and *cinnabar* on chromosome 2 in the *XXY*, triplo-4 mutant. Despite (Brown *et al.* 2020) observing decreases in canonical heterochromatic marks across the chromosome 2 pericentromere in *XXY* flies, we were surprised to observe no change in

**Table 1.** Estimated changes in repetitive DNA content from wild-type [calculated from genome and heterochromatin sizes reported in (Hoskins *et al.* 2002)] as well as CE values on chromosomes 2 and 3 for various mutants.

| Genotype | Estimated change in repetitive DNA content from WT | Chr. 2 CE | Chr. 3 CE |
|---|---|---|---|
| WT | — | 0.92 | 0.90 |
| XXY, triplo-4 | +37% | 0.90 | 0.91 |
| XXY | +35% | 0.84** | 0.85 |
| Triplo-4 | +3% | 0.89 | — |
| C(1)DX/Y | +<35% | 0.88 | — |
| Df(2R)M41A10 | −9% | — | 0.87 |
| Blm | — | 0.36*** | — |

A 2-tailed Fisher's exact test was used to calculate *P* values between observed and expected CE. **$P < 0.0017$ for chromosome 2 and $P < 0.0025$ for chromosome 3; ***$P < 0.0001$. All others were not significantly different (to account for mulitple comparisons, not significant = $P > 0.0083$ on chromosome 2 or $P > 0.0125$ on chromosome 3). Complete datasets can be found in Supplementary Tables 1 and 2. *Blm* data are obtained from Hatkevich *et al.* (2017).

centromere-proximal CO frequencies in XXY, triplo-4 flies, compared with wild type (Fig. 2a). CE for this mutant was 0.90, a non-significant difference from the wild-type CE of 0.92 for chromosome 2 (Table 1).

We also measured CO density between the same chromosome 2 markers in flies that were only XXY, without an extra copy of chromosome 4. Here too, we did not observe significant differences in proximal CO frequencies from wild type (Fig. 2c). CE for this mutant was 0.84 (Table 1), and although this was moderately significantly different from the wild-type value of 0.92 according to a 2-tailed Fisher's exact text between observed and expected proximal CO values, we do not think this is indicative of a biologically relevant decrease in centromere effect strength for 2 reasons: (1) on chromosome 2, CE for Blm syndrome helicase mutants, a positive control for centromere effect loss, is 0.36 and is extremely significantly different from wild type (Hatkevich *et al.* 2017). The XXY CE value is much closer to the wild-type value than that of *Blm* mutants, and (2) COs are strongly decreased in this mutant chromosome-wide, with this perceived difference in CE likely arising from the as-yet-unknown underlying cause of that.

While we would have liked to measure CO distribution in mutants with a broader range of increases in repetitive DNA content, it is hard to find or build fly stocks that tolerate the structural rearrangements that would make this possible. Due to this limitation, while we have data on the *trans*-effects of larger (~35–37%) increases in repetitive DNA content, we only have one karyotype with an intermediate increase in satellite DNA: C(1)DX/Y, another XXY mutant with attached X chromosomes. Although C(1)DX/Y has a Y chromosome's worth of increase in repetitive DNA when compared with XX females, the compound X chromosomes are missing large regions of pericentromeric heterochromatin, particularly the *rDNA* locus (Lindsley and Zimm 1992). C(1)DX females are always XXY as they require *rDNA* on the Y chromosome for survival and at maximum have an ~30% increase in repetitive DNA content compared with wild-type levels. When we measured CO distribution along chromosome 2 in this mutant, we once again observed no change in CE compared with wild type (Supplementary Fig. 1; Table 1).

We also measured centromere-proximal CO frequencies in triplo-4 flies, which have a low (~3%) increase in repetitive DNA content. These flies were also built using stocks with attached 4 chromosomes to ensure wild-type centromere numbers. Consistent with our previous data, CO distribution along chromosome 2 in triplo-4 flies showed no change in centromere-proximal CO frequencies (Fig. 2d) or CE value from wild type (Table 1),

suggesting that no amount of increase in total repetitive DNA content has *trans*-effects on centromere-proximal CO frequencies or the strength of the centromere effect.

We next investigated whether XXY and XXY, triplo-4 mutants have a similar lack of effect on centromere-proximal CO frequencies on other chromosomes as well. To test this, we measured CO distribution along the 57.7 Mb of chromosome 3, between recessive markers *roughoid* and *claret*. The centromere of chromosome 3 lies in the interval between markers *scarlet* and *curled* and contains about 22.8 Mb of assembled sequence. Our chromosome 3 results for XXY, triplo-4 flies followed the same pattern as on chromosome 2, with no change in proximal CO frequencies and a CE of 0.91, not significantly different from the wild-type CE value of 0.90 on chromosome 3 (Fig. 2c). XXY flies also followed chromosome 2 patterns, with a CE value of 0.85 (Supplementary Fig. 1; Table 1), mildly significantly different from the wild-type value of 0.90. However, since this number is much closer to the wild-type range than the CE of $Blm^{-/-}$ mutants on chromosome 2 had been, we are once again skeptical of assigning biological relevance to the difference observed. Overall, these results suggest that increasing total repetitive DNA content up to ~37% has no effect on centromere-proximal CO frequencies in *trans*.

## Decreases in total repetitive DNA content do not exert *trans*-effects on the CE

Last, we asked whether decreases in repetitive DNA content can affect CE strength in *trans*. While deleting large chunks of satellite DNA is difficult to do, particularly as large parts of the *D. melanogaster* pericentromere remain unassembled, we were able to use an existing mutant with a large deficiency in chromosome 2, Df(2R)M41A10. This stock has ~11 Mb of pericentromeric repetitive DNA deleted on chromosome 2 (Hilliker 1976), which is an ~9% decrease in total repetitive DNA content in heterozygotes, based on the size of *D. melanogaster* chromosomes and chromatin domains (Hoskins *et al.* 2002). We procured this mutant from the Bloomington Stock Center and genetically confirmed the deficiency by crossing to recessive mutants of homozygous lethal genes (*rolled*, *uex*, and *Nipped-B*) located within the deleted area.

As Df(2R)M41A10 is on chromosome 2 and does not live as a homozygote, we measured CO distribution along chromosome 3 in heterozygotes to test whether a decrease in repetitive DNA content influences proximal CO frequencies in *trans*. Consistent with our previous data, we observed no change in proximal CO numbers (Fig. 3) or CE (Table 1), leading us to strongly conclude that, as with centromere number, dosage changes in repetitive DNA content are unable to exert a *trans*-effect on the centromere effect.

## No *trans*-effects of satellite DNA–binding proteins on the centromere effect

On observing that satellite DNA dosage does not have *trans*-effects on CE strength, we wondered whether this was also true of satellite DNA function. Although historically thought of as genomic junk, we now know that satellite DNA is important in many ways: forming and structurally defining centromeric chromatin (Murphy and Karpen 1998), acting as a fertility barrier between species (Ferree and Barbash 2009; Jagannathan and Yamashita 2021), aiding in the pairing and segregation of achiasmate chromosomes (Dernburg *et al.* 1996), as well as facilitating the formation of chromocenters (Gall *et al.* 1971).

Recent studies have shown that chromocenter formation in *D. melanogaster* is dependent on satellite DNA–binding proteins D1 and Proliferation disrupter (Prod). D1 is an AT-hook protein that binds to the {AATAT}$_n$ satellite and is necessary for chromocenter

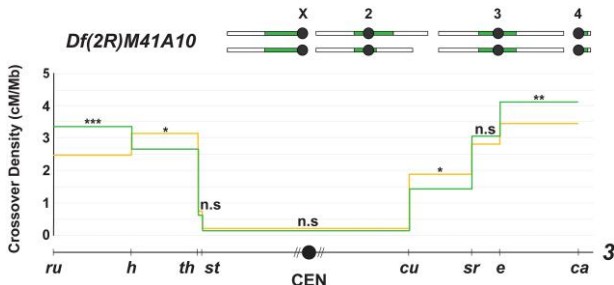

**Fig. 3.** CO distribution along chromosome 3 in *Df(2R)M41A10* flies (yellow line, *n* = 1,773) and wild-type flies (green line, *n* = 1,728), with a schematic of the *Df(2R)M41A10* karyotype shown above the graph. CO density in cM/Mb is indicated on the y-axis, and relative physical distances between markers used to score COs are indicated on the x-axis. The black circle represents the centromere, with the dashed lines around it representing the pericentromeric repetitive sequence that remains unassembled. Statistical significance in each interval was calculated using a 2-tailed Fisher's exact test between the difference in total CO vs NCO numbers in mutants from wild-type flies (ns *P* > 0.0125, \**P* < 0.0125, \*\**P* < 0.0025, \*\*\**P* < 0.00025 after correction for multiple comparisons). The complete dataset can be found in Supplementary Table 2.

formation in spermatocytes (Jagannathan *et al.* 2018), while Prod is a DNA-binding protein that binds to Prodsat {AATAACATAG}$_n$ and is necessary for chromocenter formation in imaginal discs (Jagannathan *et al.* 2019). In our study, we used mutants of *D1* and *prod* to ask whether centromere-clustering functions of satellite DNA are important for CO suppression in centromere-proximal regions.

First, we assayed whether centromere-declustering is indeed observed in the female germline of *D. melanogaster* by comparing percent declustering in meiotic cells of wild-type germaria with germaria of satellite DNA–binding protein mutants. As there are 4 pairs of homologous chromosomes in the *D. melanogaster* genome, between 1 and 3 clusters per nucleus is evidence of regular centromere-clustering, while the presence of 4 or more clusters in a single nucleus indicates declustering. We measured centromere clustering by staining wild-type, *D1*, and heterozygous *prod* mutant germaria for CENP-C (Fig. 4a) and counting the number of meiotic nuclei (as defined by the expression of synaptonemal complex protein C(3)G) that had >3 foci colocalizing with DAPI-dense regions of the cell (Fig. 4b). We observed very low percentages of centromere-declustering in wild-type as well as heterozygous *prod* mutant flies but a strongly significant (*P* < 0.0001) increase in declustering in the *D1* mutant (Fig. 4c).

To investigate whether this declustering in the *D1* mutant has downstream effects on recombination and CE strength, we measured CO frequencies along chromosome 2 in the *D1^{LL03310}/Df(3R)BSC^{666}* mutant flies (Fig. 5a). Extremely surprisingly, we once again observed no significant change in CO frequencies in the interval containing the centromere, with a CE value of 0.90, a nonsignificant change from the CE value of 0.92 on chromosome 2 in wild-type flies (Fig. 5c). These data suggest that centromere-declustering in meiotic nuclei does not influence centromere-proximal CO frequencies or centromere effect strength.

Although we did not observe declustering in meiotic cells of the *prod* mutant, we still measured CO distribution along chromosome 3 in *prod^K*/+ flies and observed no change in centromere-proximal CO frequencies or CE value compared with wild type (Fig. 5, b and c). However, it must be noted that *prod* is an essential gene and since homozygous mutants are inviable, we were limited to measuring recombination in females heterozygous for a null mutation.

Collectively, our results suggest that these satellite DNA–binding proteins do not exert a *trans*-effect in manifesting the centromere effect, through centromere-clustering or other mechanisms of action. Our results also show that centromere-clustering is not a necessary part of the mechanism through which centromere-proximal COs are suppressed, suggesting that in *D. melanogaster*, the centromere effect does not depend on satellite DNA function, a pattern consistent with what we see for satellite DNA dosage.

## Discussion

Studies on the *D. melanogaster* centromere effect have historically focused on the local, *cis*-acting contributions of the centromere and pericentromeric heterochromatin. In this study, we test whether dosage changes in certain structural components of chromosomes—centromeres and repetitive DNA—exert *trans*-effects on centromere-proximal CO suppression in *D. melanogaster*. Despite previous studies suggesting that both factors are likely to have genome-wide effects on this patterning phenomenon, our study finds that the centromere effect is surprisingly robust to dosage changes in both centromere number and the quantity of repetitive DNA. Below, we discuss the mechanistic interpretations of these findings.

### Centromere number and the CE

On reducing total centromere number using fly stocks with compound chromosomes of X and 4, we expected to see a reduction in centromere-proximal CO frequencies, based on Redfield's observation of increased proximal COs in triploids (that have 24 centromeres compared with 16 in diploids; Redfield 1930, 1932). This would suggest that centromeres can act as molecular sinks to a putative "centromere effect factor" necessary to maintain the wild-type levels of centromere-proximal CO suppression, with a dosage reduction in centromere number leading to increased availability of this "centromere effect factor," potentially leading to a strengthened centromere effect. Alternatively, increased centromere-proximal CO in flies with a reduced number of centromeres would suggest that a baseline centromere number is required to maintain the centromere effect at wild-type levels. With centromere clustering observed during meiotic prophase I when recombination occurs (Hatkevich *et al.* 2021), this would potentially have indicated a role for the chromocenter in establishing the centromere effect.

However, in our study, we observed no change in centromere effect strength when total centromere number is reduced, suggesting that centromeres are neither acting as molecular sinks nor spatially regulating centromere-proximal CO frequencies within the chromocenter. This deviation from what we expected based on Redfield's data suggests that the increase observed in her study is likely not due to changes in total centromere number and perhaps a result of ploidy changes in triploids. It is also possible that changing centromere numbers could only exert *cis*-effects on centromere-proximal COs, a hypothesis that is hard to test as mapping recombination within compound chromosomes is near impossible. Another possibility is that reducing the total centromere number by 4 is not sufficient to cause an effect. To reduce centromere numbers further, we would need mutants with an even greater number of attached chromosomes. Unfortunately, flies heterozygous for markers in this background are highly difficult to generate, if not inviable.

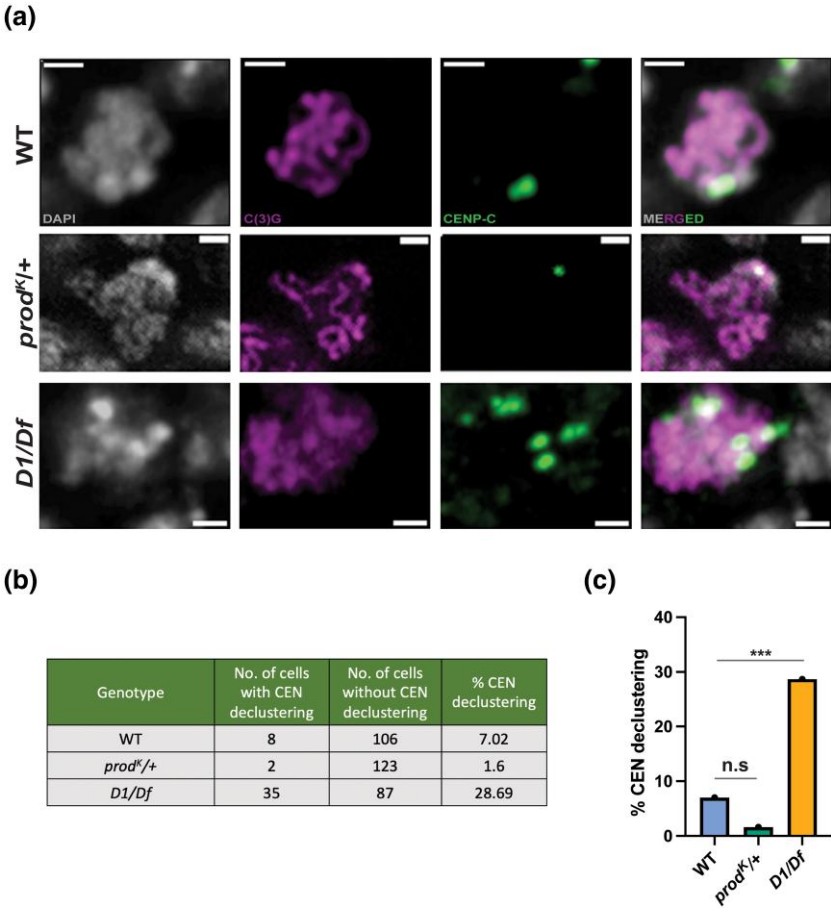

**Fig. 4.** a) Representative images of meiotic nuclei [defined as nuclei expressing synaptonemal complex protein C(3)G (magenta)] showing 2, 1, and 6 CENP-C foci (green) that colocalize with DAPI-dense nuclear regions (bright gray puncta) in wild-type (top panel), $prod^K$/+ (middle panel), and $D1^{LL03310}$/$Df(3R)BSC^{666}$ (bottom panel) flies, respectively. The scale bars correspond to 1 μm. b) A table showing the number of meiotic nuclei with and without centromere-declustering (defined as >3 CENP-C foci colocalizing with DAPI-dense regions/nucleus) in regions 2A and 2B of the germarium, as well as centromere-declustering percentages in wild-type, $prod^K$/+, and $D1^{LL03310}$/$Df(3R)BSC^{666}$ flies. c) A bar graph showing the percentage of centromere-declustering in meiotic cells from wild-type ($n=119$), $prod^K$/+ ($n=127$), and $D1^{LL03310}$/$Df(3R)BSC^{666}$ ($n=127$) flies. A 2-tailed Fisher's test was used to calculate the significance between the number of meiotic cells with and without centromere-declustering across genotypes, with nonsignificant ($P=0.0534$) differences observed between wild-type and $prod^K$/+ flies and very significant ($P<0.0001$) differences observed between wild-type and $D1^{LL03310}$/$Df(3R)BSC^{666}$ flies.

## Repetitive DNA content and the centromere effect

Westphal and Reuter (2002) demonstrated that the mutants of *Su(var)* genes coding for important heterochromatin factors such as HP1 and H3K9 methyltransferases displayed an increase in centromere-proximal CO frequencies. Combined with extensive evidence that extra copies of chromosome Y affect pericentromeric heterochromatinization genome-wide in *D. melanogaster* (Dimitri and Pisano 1989; Brown *et al.* 2020; Delanoue *et al.* 2023), we hypothesized that centromere-proximal CO frequencies would be increased in *XXY* flies and other mutants with dosage changes in repetitive DNA content. This would be consistent with extra repetitive DNA soaking up factors necessary to establish and maintain pericentromeric heterochromatin or the CE, leading to decreased availability in the rest of the genome, potentially allowing for increased centromere-proximal COs. Alternatively, increased repetitive DNA may have led to a stronger centromere effect, which would suggest a more direct role for repetitive DNA in maintaining CO suppression around the centromere.

In our study, we observed no change in centromere-proximal CO frequencies with dosage changes in repetitive DNA, suggesting that the loss of heterochromatinization observed in *XXY* flies by

Brown *et al.* (2020) is not sufficient to allow for increased centromere-proximal COs. The increase in proximal COs observed by Westphal and Reuter (2002) in certain *Su(var)* mutants with defective heterochromatinization suggests that there is a threshold of heterochromatin loss necessary for an increase in CO frequencies; if true, our study indicates that in *XXY*, triplo-4 flies, a 37% increase in repetitive DNA content does not reach this threshold. Interestingly, the phenomenon of position effect variegation in *XXY* flies suggests that the threshold of heterochromatic disruption necessary for changes in gene expression is lower than that necessary for recombination and that the open chromatin landscape in these mutant flies can be permissive to transcriptional machinery without being permissive to DSB or recombination machinery. Additionally, the lack of change observed in our study also rules out spatial regulation of the centromere effect by repetitive DNA—potentially within structures such as the chromocenter—as dosage changes in repetitive DNA content were not directly proportional to centromere effect strength.

## Repetitive DNA function and the CE

As our study demonstrates that the *D. melanogaster* centromere effect is unaffected by dosage changes in repetitive DNA, we

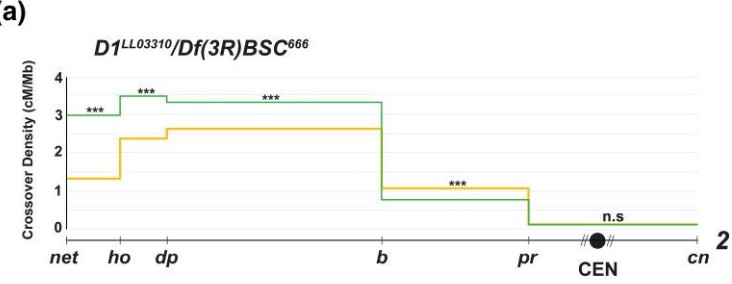

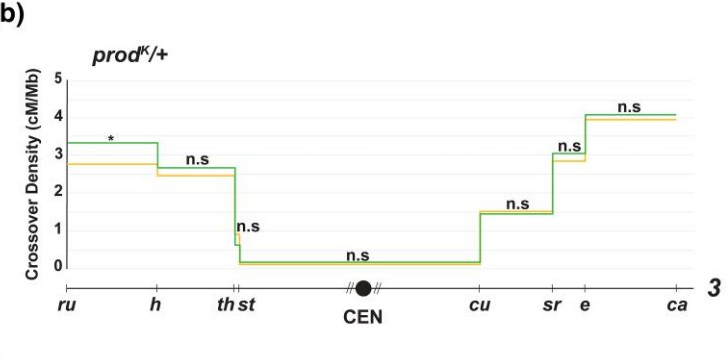

(c)

| Genotype | Chr. *2* CE value | Chr. *3* CE value |
|---|---|---|
| WT | 0.92 | 0.90 |
| D1$^{LL03310}$/Df(3R)BSC$^{666}$ | 0.90 | - |
| prod$^K$/+ | - | 0.91 |

**Fig. 5.** a) CO distribution along chromosome 2 in $D1^{LL03310}/Df(3R)BSC^{666}$ flies (yellow line, $n = 6,399$) and wild-type flies (green line, $n = 4,104$). CO density in cM/Mb is indicated on the y-axis, and relative physical distances between markers used to score COs are indicated on the x-axis. The black circle represents the centromere, with the dashed lines around it representing the pericentromeric repetitive sequence that remains unassembled. Statistical significance in each interval was calculated using a 2-tailed Fisher's exact test between the difference in total CO vs NCO numbers in mutants from wild-type flies (ns $P > 0.0083$, \*$P < 0.0083$, \*\*$P < 0.0017$, \*\*\*$P < 0.00017$ after correction for multiple comparisons). The complete dataset can be found in Supplementary Table 1. b) CO distribution along chromosome 3 in $prod^K$/+ flies (yellow line, $n = 5,320$) and wild-type flies (green line, $n = 1,728$). CO density in cM/Mb is indicated on the y-axis, and relative physical distances between markers used to score COs are indicated on the x-axis. The black circle represents the centromere, with the dashed lines around it representing the pericentromeric repetitive sequence that remains unassembled. Statistical significance in each interval was calculated using a 2-tailed Fisher's exact test between the difference in total CO vs NCO numbers in mutants from wild-type flies (ns $P > 0.0125$, \*$P < 0.0125$, \*\*$P < 0.0025$, \*\*\*$P < 0.00025$ after correction for multiple comparisons). The complete dataset can be found in Supplementary Table 2. c) A table containing CE values on chromosomes 2 and 3 of $D1^{LL03310}/Df(3R)BSC^{666}$ and $prod^K$/+ flies, respectively. The 2-tailed Fisher's test was used to calculate the significance between observed and expected proximal CO values and was nonsignificantly different from wild type for both mutants.

wondered whether disrupting repetitive DNA function using satellite DNA–binding protein mutants would have genome-wide effects on the CE. To test this, we measured centromere-proximal CO frequencies in mutants that lack the AT-hook protein D1, which binds to the {AATAT}$_n$ satellite and has been shown by us and Jagannathan et al. (2018) to cause declustering in fly oocytes and spermatocytes, respectively. Surprisingly, we once again observed no change in centromere effect strength in this or a *prod* mutant, suggesting not only that centromere-declustering does not play a role in establishing the CE in *D. melanogaster* but also that satellite DNA–binding proteins do not influence the centromere effect via other mechanisms either.

It is noteworthy, however, that the {AATAT}$_n$ satellite is primarily found on the *X* chromosome, making it possible that centromere effect disruption in D1 mutants takes place only in *cis*.

This can be tested by measuring centromere effect strength on chromosome *X* in these mutants, although the centromere effect is already very weak on this chromosome, likely due to the expanse of satellite DNA on it. Despite this caveat, our results do indeed demonstrate that the {AATAT}$_n$ satellite–binding protein D1 does not have global roles in establishing the *D. melanogaster* centromere effect.

## Centromere effect mechanism

Our study rules out the mechanistic role of structural chromosomal components such as centromeres and repetitive DNA in suppressing centromere-proximal COs in *trans*, suggesting that the *D. melanogaster* centromere effect is more likely to be genetically than spatially controlled. Although previous studies from our lab have shown that CO interference and assurance are genetically

separable from the centromere effect (Brady *et al.* 2018), the emergence of interference models such as coarsening makes us question whether the centromere effect could also be explained by variations of these models. Post-translational modifications of pro-CO meiotic proteins are being heavily investigated for their role in establishing CO interference (Zhang *et al.* 2021; Haversat *et al.* 2022), and a speculative idea we would like to put forth is that such modifications could also be manifesting the centromere effect through modifying enzymes (like kinases) being anchored at the centromere and biasing neighboring chromosomal regions—less-repetitive beta heterochromatin and proximal euchromatin—toward NCO repair. While the factors involved may be different from those in CO interference, such a model sees both patterning phenomena manifesting through similar modes of genetic control, an idea that is further supported by our study ruling out structural and spatial contributions to centromere-proximal CO suppression.

## Conclusion

Our study shows that the centromere effect is robust not only to dosage changes in various structural components of chromosomes, such as centromeres and repetitive DNA, but also to changes in certain aspects of satellite DNA function, such as centromere clustering. These results strongly suggest that the centromere effect is likely not controlled spatially during meiotic prophase and is perhaps mediated through genetic factors, opening up avenues of future research to uncover the exact mechanistic details of centromere-proximal CO suppression in *D. melanogaster*.

## Data availability

*Drosophila* stocks are available upon request. The authors confirm that all data necessary for confirming the conclusions of the article are present within the article, figures, table, and Supplementary material online.

Supplemental material available at GENETICS online.

## Acknowledgments

We thank Yukiko Yamashita for the fly stocks that were generously sent to us, Yves Barral for the suggestion that a centromere-anchored kinase might contribute to the centromere effect, Kim McKim for the CENP-C antibody, Nicole Crown for the C(3)G antibody, and members of the Sekelsky lab for helpful comments on the manuscript.

## Funding

This work was supported by a grant from the National Institute of General Medical Sciences to JS under award 1R35GM118127. NMP was supported in part by a grant from the National Institute on Aging under award 1F31AG079626-01 and from the National Institute of General Medical Sciences under award T32GM135128. LKF was supported in part by a grant from the  Division of Biological Infrastructure under award 2048087.

## Conflicts of interest

The author(s) declare no conflicts of interest.

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
