## [Peer Review File · Genetics]

Centromere-Proximal Suppression of Meiotic Crossovers in *Drosophila* is Robust to Changes in Centromere Number, Repetitive DNA Content, and Centromere Clustering

Nila Pazhayam, Leah Frazier, and Jeff Sekelsky

NOTE: The reviews and decision letters are unedited and appear as submitted by the reviewers.

In extremely rare instances and as determined by a Senior Editor or the EIC, portions of a review may be redacted. If a review is signed, the reviewer has agreed to no longer remain anonymous.

The review history appears in chronological order.

Review Timeline:

Submission Date:	2023-10-16
Editorial Decision:	2023-11-12
Resubmission Received:	2023-12-05
Accepted:	2023-12-10

November 12, 2023

GENETICS-2023-306564

Centromere-Proximal Suppression of Meiotic Crossovers in *Drosophila* is Robust to Changes in Centromere Number and Repetitive DNA Content

Dear Dr. Sekelsky:

Two experts in the field have reviewed your manuscript, and I have read it as well. I am pleased to inform you that, with minor revisions, it is potentially suitable for publication in GENETICS.

Both reviewers felt that the experiments were well thought out and the data were readily interpretable. Although the data are essentially negative throughout, there was prior evidence indicating that these experiments would uncover a mechanism. The fact no phenotype was uncovered supports that these aspects of chromosome biology are not involved in the centromere effect, which is an important result on its own.

Both reviewers have comments and concerns that need to be addressed in a revised manuscript. You can read them at the end of this email. If possible, please strongly consider the experiment suggested by Reviewer #1 regarding centromere clustering in ProdK/+ females (and presumably D1 females) - this would be useful information for the field and would help to interpret the data from your experiment. Other suggested experiments would be welcome, but would not be necessary for the paper to be published.

We look forward to receiving your revised manuscript. Please let the editorial office know approximately how long you expect to need for revisions.

Upon resubmission, please include:

1. A clean version of your manuscript;
2. A marked version of your manuscript in which you highlight significant revisions carried out in response to the major points raised by the editor/reviewers (track changes is acceptable if preferred);
3. A detailed response to the editor's/reviewers' comments and to the concerns listed above. Please reference line numbers in this response to aid the editors.

Additionally, please ensure that your resubmission is formatted for GENETICS.

<https://academic.oup.com/genetics/pages/general-instructions>

Follow this link to submit the revised manuscript: Link Not Available

Sincerely,

Jack Bateman
Associate Editor
GENETICS

Approved by:
Howard Lipshitz
Editor in Chief
GENETICS

Reviewer #1 (Comments for the Authors (Required)):

Pazhayam et al. attempted to address a fundamental question regarding the influence of centromeres and repetitive DNA on the centromere effect (the absence of crossovers in the pericentromeric heterochromatin). The centromere effect has been investigated in *Drosophila* in previous work through different experimental approaches, but the mechanisms behind this process are still unknown. There was previous evidence suggesting centromere dosage and repetitive DNA play a role in the centromere effect. The authors extensively tested the importance of centromere and repetitive DNA dosage and unfortunately found crossovers were unchanged in any of the conditions they tested. This has been an open question in the field for a very long time and the work presented here (while negative data) is still an important step forward in increasing our understanding of the

mechanisms behind the centromere effect. Additionally, the authors experiments were very thorough and I don't have any experimental improvements other than point 1. Overall, the paper was well written, clear, and easy to follow. I have put my comments below.

Comments:

1. Line 391-393: I was unclear if the authors know what centromere clustering looks like in oocytes in *ProdK/+* females. I looked at the original paper from Jagannathan et al. and did not see oocyte data. If the authors don't know what centromere clustering looks like in females this is an experiment they could do. It is easy to examine centromere clustering via antibody staining. They did address the possibility that the phenotype in females might not be the same but it would be a more complete experiment if they are confident there is a clustering phenotype in oocytes in those mutants.

2. Line 133-135: "it is unclear whether this effect is due to a change in total centromere number from 16 to (one per sister chromatid for eight chromosomes) to 24 in triploids (one per sister chromatid for twelve chromosomes)"

This description was confusing- the authors could rewrite this to be clearer. Also, the "to" is repeated.

3. Line 154-158: The sentences about recombination intervals should be rewritten to be more accessible to a non-*Drosophila* audience. It could be clearer.

4. Line 160: The genotype *C(1)/0 ; C(4)/0* should be referenced earlier (probably at line 149) to make it clear what the genotype is for the compound X and 4 for readers not familiar with *Drosophila* genotypes.

5. Line 194-196: The descriptions of the increase in repetitive DNA was a bit confusing. I would recommend outlining the XXY and triplo-4 flies prior to the double.

Reviewer #2 (Comments for the Authors (Required)):

This paper investigates a widespread phenomenon in eukaryote genomes - specifically the suppression of meiotic crossovers in proximity to centromeres - taking a genetic approach in *Drosophila*. Specifically, they alter the dosage of centromeres, and repeat DNA content, and analyse how this influences crossovers.

First the authors use *Drosophila* stocks with compound X and Chr4, creating dicentric (at the sequence level) chromosomes. The authors use CENP-C staining to show that the fusion chromosomes only have a single centromere focus - this is interesting and shows that one of the centromeres has epigenetically inactivated. Is it always the same centromere which inactivates? Is the remaining centromere in its normal position? It might be interesting to perform CENP-A or CENP-C ChIP-seq in these lines to provide a higher resolution map of centromere location in these strains?

Using these lines the authors then map crossovers on chromosome 2 across the centromere. Using segregation of phenotypic markers, no difference is observed on chromosome 2, in the presence of the X-Chr4 fusion.

Map distances are compared to physical distances based on genome assembly 6.53 - is this assembly known to be complete through the centromeres?

Next the authors analysed XXY triplo-4 flies and mapped crossovers on chromosomes 2 and 3, but only weak effects observed. I was wondering where the piRNA clusters are located in *Drosophila*? Are they located in crossover suppressed pericentromere intervals? I am aware that piRNA clusters are highly polymorphic between strains - perhaps that could be a useful way to address the question of varying heterochromatin on the crossover landscape. I believe recent studies have also used Cas9 to delete the piRNA clusters, which could also be useful? In terms of trans effects, maybe using known heterochromatic mutants might also be productive - eg *suv3-9*? Along these lines, the authors test the D1 and *prod* mutants, which encode proteins known to bind satellite repeats, but again no effects were seen.

In summary, the authors attempt a number of experiments to see whether the strength of the CE can be modified, but without any significant changes. What is known of the structural differences between the centromeres of the strains used? Could extensive SVs within the centromeres be causing crossover suppression, irrespective of whatever else is changing? The work overall is well performed and analysed appropriately - however, I think the authors could make more use of the available complete centromere assemblies available in *Drosophila*.

Minor points: Line 37 references Beadle 1932 for the CE - however, this effect is widespread and seen in diverse plant, animals and fungi. I think a more extensive set of references showing this effect is common across species, and across different types of centromeres, is worthwhile (ie point, regional, satellite, RT-based).

Line 57-59: Im not sure it is accurate that *D.melanogaster* centromeres are satellite based. I note recent work that shows in fact many of the CENP-A occupied regions in *Drosophila* species are retrotransposon based :
<https://www.biorxiv.org/content/10.1101/2023.08.22.554357v1>

Reviewer #1 (Comments for the Authors (Required)):

Pazhayam et al. attempted to address a fundamental question regarding the influence of centromeres and repetitive DNA on the centromere effect (the absence of crossovers in the pericentromeric heterochromatin). The centromere effect has been investigated in *Drosophila* in previous work through different experimental approaches, but the mechanisms behind this process are still unknown. There was previous evidence suggesting centromere dosage and repetitive DNA play a role in the centromere effect. The authors extensively tested the importance of centromere and repetitive DNA dosage and unfortunately found crossovers were unchanged in any of the conditions they tested. This has been an open question in the field for a very long time and the work presented here (while negative data) is still an important step forward in increasing our understanding of the mechanisms behind the centromere effect. Additionally, the authors experiments were very thorough and I don't have any experimental improvements other than point 1. Overall, the paper was well written, clear, and easy to follow. I have put my comments below.

Comments:

1. Line 391-393: I was unclear if the authors know what centromere clustering looks like in oocytes in *ProdK/+* females. I looked at the original paper from Jagannathan et al. and did not see oocyte data. If the authors don't know what centromere clustering looks like in females this is an experiment they could do. It is easy to examine centromere clustering via antibody staining. They did address the possibility that the phenotype in females might not be the same but it would be a more complete experiment if they are confident there is a clustering phenotype in oocytes in those mutants.

We thank the reviewer for this suggestion and have included this experiment in this new version of the manuscript. While the *ProdK/+* females do not show centromere-declustering in oocytes, *D1/Df* oocytes show significant declustering, making the lack of effect on the CE in this mutant even more striking. This result has been included as Figure 4 and discussed between lines 292 - 334. Its implications have been referenced in the title, abstract (lines 13 - 14), introduction (lines 118 - 119, 122 - 125), discussion (lines 410 - 414) and conclusions (lines 443 - 444).

2. Line 133-135: "it is unclear whether this effect is due to a change in total centromere number from 16 to (one per sister chromatid for eight chromosomes) to 24 in triploids (one per sister chromatid for twelve chromosomes)" This description was confusing- the authors could rewrite this to be clearer. Also, the "to" is repeated.

We have re-written this line to make it clearer (lines 137 - 138).

3. Line 154-158: The sentences about recombination intervals should be rewritten to be more accessible to a non-*Drosophila* audience. It could be clearer.

We have re-written this paragraph to make it clearer (lines 158 - 162).

4. Line 160: The genotype $C(1)/0$; $C(4)/0$ should be referenced earlier (probably at line 149) to make it clear what the genotype is for the compound X and 4 for readers not familiar with *Drosophila* genotypes.

We now reference the $C(1)/0$; $C(4)/0$ genotype where we first mention a stock with compound X and 4th chromosomes (line 143).

5. Line 194-196: The descriptions of the increase in repetitive DNA was a bit confusing. I would recommend outlining the XXY and triplo-4 flies prior to the double.

To make this easier to understand for readers, we have re-written this paragraph to outline mutants in the following order: 1) triplo-4, 2) XXY, 3) XXY, triplo-4 (lines 199 - 204).

Reviewer #2 (Comments for the Authors (Required)):

This paper investigates a widespread phenomenon in eukaryote genomes - specifically the suppression of meiotic crossovers in proximity to centromeres - taking a genetic approach in *Drosophila*. Specifically, they alter the dosage of centromeres, and repeat DNA content, and analyse how this influences crossovers.

First the authors use *Drosophila* stocks with compound X and Chr4, creating dicentric (at the sequence level) chromosomes. The authors use CENP-C staining to show that the fusion chromosomes only have a single centromere focus - this is interesting and shows that one of the centromeres has epigenetically inactivated.

1. Is it always the same centromere which inactivates? Is the remaining centromere in its normal position? It might be interesting to perform CENP-A or CENP-C ChIP-seq in these lines to provide a higher resolution map of centromere location in these strains?

Most of these compound chromosomes were made (sometimes by irradiation) or discovered in stocks mostly in the first half of the 20th century. They have now been kept in live stocks for many decades. As the centromeres of these compound stocks haven't been sequenced, we do not know whether the single CENP-C focus found on a compound chromosome is due to epigenetic inactivation, or there being only one centromeric sequence. CENP-A ChIP-seq might tell us whether there are two centromeres (defined via DNA sequence) with only one of them remaining active, or if there is just one copy of centromeric DNA in these compound chromosomes. While this is beyond the scope of our current work, it is indeed a very interesting question to ask (and answer).

Using these lines the authors then map crossovers on chromosome 2 across the centromere. Using segregation of phenotypic markers, no difference is observed on chromosome 2, in the presence of the X-Chr4 fusion.

2. Map distances are compared to physical distances based on genome assembly 6.53 - is this assembly known to be complete through the centromeres?

While centromere sequences are known from Chang et al., 2019, sequence assembly isn't complete for large satellite arrays within pericentromeric heterochromatin, due to their highly repetitive nature. However, various sequencing techniques [Hoskins et al. 2002, Adams et al. 2000, Khost et al. 2017] as well as quantification of *in situ* hybridization probes to these repetitive regions [Lohe et al., 1993] have provided approximate physical sizes of many satellite arrays as well as the total size of pericentromeric heterochromatin of each arm in *D. melanogaster*.

Next the authors analysed XXY triplo-4 flies and mapped crossovers on chromosomes 2 and 3, but only weak effects observed.

3. I was wondering where the piRNA clusters are located in *Drosophila*? Are they located in crossover suppressed pericentromere intervals? I am aware that piRNA clusters are highly polymorphic between strains - perhaps that could be a useful way to address the question of varying heterochromatin on the crossover landscape. I believe recent studies have also used Cas9 to delete the piRNA clusters, which could also be useful?

The piRNA clusters in *D. melanogaster* are indeed located in the less-repetitive beta-heterochromatic regions of the pericentromere where COs are suppressed. However, since their sizes are estimated in the scale of kilobases (up to a few hundred), we don't expect deletions in piRNA clusters to have an effect on recombination landscape, especially since a deletion of ~9 Mb of repetitive DNA in the *Df(2R)M41A10* flies did not affect the strength of the CE. It is also possible that deletion of piRNA clusters may lead to increased mobilization of pericentromeric TEs, which may lead to increased proximal mitotic crossovers that are can't be distinguished from meiotic crossovers (there will not be "jackpots" unless they occur in stem cells; in the female germline each stem cell division produces a single oocyte).

4. In terms of trans effects, maybe using known heterochromatic mutants might also be productive - eg *suv3-9*?

Su(var) mutants - particularly *Su(var)3-9*, *Su(var)3-7*, and *Su(var)2-5* - would indeed be very interesting to study the *trans* effects of heterochromatin loss in, and we are currently in the process of looking at centromere-proximal recombination in these mutants as a part of another study! We have hypothesized that these may impact the part of the centromere effect that results from the absence of DSBs in alpha heterochromatin, which we believe to be different from the effect we set out to examine here, which is reduced crossovers in regions that do receive DSBs.

Along these lines, the authors test the D1 and prod mutants, which encode proteins known to bind satellite repeats, but again no effects were seen.

In summary, the authors attempt a number of experiments to see whether the strength of the CE can be modified, but without any significant changes.

5. What is known of the structural differences between the centromeres of the strains used? Could extensive SVs within the centromeres be causing crossover suppression, irrespective of whatever else is changing? The work overall is well performed and analysed appropriately - however, I think the authors could make more use of the available complete centromere assemblies available in *Drosophila*.

While it is indeed possible that large structural variations within centromeres could be causing centromere-proximal CO suppression irrespective of other changes, *D. melanogaster* centromeres have currently only been sequenced in a few wild-type strains, making it difficult for us to comment with authority on this question. Unpublished data from the Larracuente lab has shown that centromeric sequences can vary even within the same wild-type strains kept in different labs, suggesting that the DNA sequence definition of centromeres may not be uniform even within a species. While this is very interesting and may have fascinating implications on evolution and recombination, investigating the role of centromeric sequence/structural variations in recombination is a huge endeavor and would constitute a whole other project.

It is also possible that variations in satellite sequences or TE arrays in beta heterochromatin may contribute to suppression. In our previous study (Hartmann *et al.* 2019) we showed that the suppression extends for megabases into the euchromatin, so it cannot all be due to structural heterozygosity.

6. Minor points: Line 37 references Beadle 1932 for the CE - however, this effect is widespread and seen in diverse plant, animals and fungi. I think a more extensive set of references showing this effect is common across species, and across different types of centromeres, is worth taking (ie point, regional, satellite, RT-based).

We have added references to centromere-proximal CO suppression in other species – Arabidopsis, budding yeast, fission yeast, maize, mammals, etc., on lines 55 - 56.

7. Line 57-59: I'm not sure it is accurate that *D. melanogaster* centromeres are satellite based. I note recent work that shows in fact many of the CENP-A occupied regions in *Drosophila* species are retrotransposon based : <https://www.biorxiv.org/content/10.1101/2023.08.22.554357v1>

The reviewer is right - while *D. melanogaster* centromeres contain some satellite sequence, they do indeed consist primarily of retrotransposons. We did not mean to say or imply that the centromeres themselves are satellite based, but that the pericentromeric heterochromatin surrounding the centromere consists of large satellite arrays in flies (and other organisms). We have re-worded this line to make it clearer (lines 59 - 60).

December 10, 2023

RE: GENETICS-2023-306700

Dr. Jeff Sekelsky
University of North Carolina at Chapel Hill
Department of Biology
CB #3280, 303 Fordham Hall
Chapel Hill, North Carolina 27599-3280

Dear Dr. Sekelsky:

Congratulations! We are delighted to inform you that your manuscript entitled "**Centromere-Proximal Suppression of Meiotic Crossovers in *Drosophila* is Robust to Changes in Centromere Number, Repetitive DNA Content, and Centromere Clustering**" is acceptable for publication in GENETICS. Many thanks for submitting your research to the journal.

When preparing your final files, please consider one minor edit: in your new paragraph describing the clustering analysis (lines 310-315), please clarify that the ProdK mutants are in fact heterozygous ProdK/+. This is clear in Figure 4 and is later clarified in the text in lines 326-328, but in the paragraph in question one may read the text to mean that it is a complete loss of ProdK function.

To Proceed to Production:

1. Format your article according to GENETICS style, as discussed at <https://academic.oup.com/genetics/pages/general-instructions>, and upload your final files at <https://genetics.msubmit.net>.
2. Your manuscript will be published as-is (unedited-as submitted, reviewed, and accepted) at the GENETICS website as an Advanced Access article and deposited into PubMed shortly after receipt of source files and the completed license to publish. Please notify sourcefiles@thegsajournals.org if you do not wish to publish your article via Advanced Access.
3. We invite you to submit an original color figure related to your paper for consideration as cover art. Please email your submission to the editorial office or upload it with your final files. You can submit a small-sized image for evaluation, and if selected, the final image must be a TIFF file 2513px wide by 3263px high (8.375 by 10.875 inches; resolution of 600ppi). Please avoid graphs and small type.

If you have any questions or encounter any problems while uploading your accepted manuscript files, please email the editorial office at sourcefiles@thegsajournals.org.

Sincerely,

Jack Bateman
Associate Editor
GENETICS

Approved by:
Howard Lipshitz
Editor in Chief
GENETICS

note: Please add jnls.author.support@oup.com and genetics.oup@kwglobal.com (or the domains @oup.com and @kwglobal.com) to your email program's "safe senders" list. You will be contacted by both at various points during the production process.